# On the Design of Composite Patch Repair for Strengthening of Marine Plates Subjected to Compressive Loads

Nikos Kallitsis and Konstantinos N. Anyfantis *

Shipbuilding Technology Laboratory, School of Naval Architecture and Marine Engineering, National Technical University of Athens, Zografos, 15780 Athens, Greece
* Correspondence: kanyf@naval.ntua.gr; Tel.: +30-210-772-1325

**Abstract:** Marine structures are susceptible to corrosion that accelerates material wastage. This phenomenon could lead to thickness reduction to the extent in which local buckling instabilities may occur. The majority of existing repair techniques require welding, which is a restricting factor in flammable environments where hot work is prohibited. A novel repair methodology that has attracted the research focus for over two decades is the adhesive bonding of a composite patch on a ship's damaged plating. Although most studies have been focused on patch repair against crack propagation, restoring the initial buckling strength of corroded marine plates is of high interest. In this work, this technique is assessed using numerical experimentation through finite element analysis (FEA) with the patch's dimensions as design parameters. The results are then evaluated using a design-of-experiments (DOE) approach by generating a response surface from central composite design (CCD) points. Applying this methodology to various plates and patches makes it possible to create a repair design procedure that specifies the minimum patch requirements depending on the metal substrate's dimensions and corrosion realized.

**Keywords:** repair; buckling; finite element analysis; composite materials; design-of-experiments; computational mechanics; ship

## 1. Introduction

Ships and offshore platforms are large marine structures operating in a saline environment. They are composed of large, welded plates that comprise the structure's main (e.g., upper deck, side shell, bulkheads, inner bottom, etc.) or secondary (e.g., walls) boundaries/dividers. The material used for the majority of these plates is marine-grade structural steel, with grades 'A' (mild steel) and 'AH' (higher tensile steel) most commonly used. Due to their large span, ships are reinforced with stiffeners such as flat bars, I, T, and L beams. Therefore, marine structures are a combination of stiffened panels subjected to external and internal loads arising, for instance, from ship motion, from green water, from sloshing, etc. Their stiffening system (transverse or longitudinal), stiffeners' cross-sections, and plates' dimensions are designed to ensure adequate strength against these loads. As a result, large strains and, subsequently, stresses are avoided, ensuring safe operation. However, the structure's strength weakens as time goes by, mainly due to material/structural failures, particularly material wastage.

Material wastage is a phenomenon that occurs from accumulated corrosion to one or both of a plate's sides, depending on whether a corrosive environment is existent. There are different types of corrosion, namely general (uniform), grooving (near welds caused by galvanic current), and pitting (random with local coating breakdown) (IACS [1]). The factors that initiate and accelerate material wastage at the structure's exterior are saline water (seawater) and heavy scale buildup. However, corrosion is not solely found externally but also internally in flooded seawater areas, such as ballast tanks, and often in cargo areas due to the hazardous chemical composition the cargo might have or due to water remaining.

These circumstances lessen the structural members' strength and could lead to unwanted results such as crack initiation/propagation, buckling, or even the material succumbing to the applied load.

The International Association of Classification Societies (IACS) has issued several guidelines aiming the surveyors' attention at regions known to be susceptible to corrosion, where thickness measurements are conducted (IACS [1]). If material wastage exists and the measurement is below the thickness's renewal value, standard practice is cropping and renewal of the plate with a thickness equal to or greater than the original. An alternative is the installation of additional stiffeners or plate doublers. However, these practices could pose a safety threat where hot work is prohibited, for instance, due to the region's flammability. Such areas could be the cargo hold or fuel tanks, where proper preparation (degassing) should be conducted for hot work to follow. Other instances are plates in areas that are not considered flammable but are neighboring with such regions (e.g., plates that separate the engine room and the fuel tank, a ballast tank and the cargo tank, or the upper deck and the cargo tank). Rules are stricter in FPSOs and offshore platforms that have a higher hazard risk. In all the above cases, the operation might need to pause for proper preparation to be conducted.

An alternative recommended repair practice proposed by Classification Societies such as Bureau Veritas (BV) and Det Norske Veritas (DNV) is the introduction of a fiber reinforced composite patch to restore the structure's integrity (BV [2] and DNV [3]). For instance, according to DNV, a bonded patch repair can be used for non-critical elements and non-critical damage in critical elements. This patch is adhesively bonded on the damaged steel plate to compensate for the thickness reduction caused by corrosion or to repair a crack (Hashim et al. [4]). However, this method does not apply to all cases, as each one should be evaluated separately based on the technique's efficiency and safety. This case-by-case assessment results from the method's unreliability due to its short span of on-site practices in the industry. Therefore, although promising, the lack of service reports constrains it, making its approval as a common repair practice difficult. For this reason, this procedure is yet to be approved by IMO for its application in the rehabilitation of load-carrying primary members.

Considering the interest behind this innovative repair technique, this work serves as additional evidence that such a method can, theoretically, serve its purpose for rehabilitating against buckling. Some indicative studies towards this objective are by Hashim et al. [4], Karatzas et al. [5], Turan [6], and Anyfantis [7]. Additionally, composites have also been investigated in larger scale applications, such as riser configurations (Salama et al. [8], Brown [9]), weighing their advantages and disadvantages against common materials. Finally, the composites' properties in a marine environment have been examined by Echtermeyer [10,11].

As aforementioned, a marine structure is composed of stiffened panels. The panels' areas between adjacent stiffeners can be considered as isolated plates with the loads being transferred between plates at their boundary edges. A typical load causes tension and compression at two parallel edges of these plates arising from hull girder primary bending. For example, the latter can occur at a ship's hogging state, where tensile stresses develop at its upper part and compressive at its lower. Although its initial design can withstand such strains, when the structural members' effective thickness is reduced due to corrosion, buckling may occur during compression.

This preliminary study undertakes the case of rehabilitating the buckling strength of a corroded plate subjected to uniform uniaxial compression through the application of a composite patch. The main benefit of using CFRP patches is the safety it provides over hot work in areas where proper degassing and scrubbing must be conducted beforehand (i.e., FPSOs, fuel tanks, cargo tanks). Thus, with the introduction of cold working, less preparation time and operational downtime are required. Additionally, as aforementioned, material wastage can occur in three ways due to corrosion: uniform, grooving, pitting. Non-uniform corrosion combined with local material imperfections that may exist could lead to uneven surfaces and structural behavior. However, since this work's scope is preliminary,

the uniform corrosion assumption and elastic structural behavior could be considered as a conservative approach to the rehabilitation method's assessment. It should be noted that the corrosion is applied to the net scantlings (i.e., design scantlings) of the structural member since the load-bearing capabilities are calculated using this value. Finally, the composite patch's exposure to water may cause hygroscopic swelling ([12–14]), which could decrease the mechanical strength of the polymeric matrix. Nevertheless, following the study's preliminary nature, it could be assumed that the repair is temporary, thus, minimizing the effects of hygroscopic swelling.

The purpose of the study is not to numerically showcase that bonding a composite patch to a metal plate increases the latter's strength. Instead, the focus is on the introduction of design guidelines when such repair practice is chosen to be applied to a plate with the risk of compressive buckling. The problem's design parameters are related to the patch's dimensions, while all other values are out of scope. Its assessment is based on elastic bucking finite element analysis (FEA) corrected for plasticity. The results are then processed using statistical methods from design-of-experiments (DoE) to help understand the problem and the effect of the design parameters on the compressive strength of the repaired plate.

## 2. Problem Set-Up

### 2.1. Background

Classification Societies' rules have issued guidelines for correctly constructing an FEA model along with the load application procedures. These guidelines are based on a global or local scale, depending on whether a case of three cargo hold, or a more specific region, such as the corners of a bulkhead's lower stool, is examined. If an area's plates' thickness measurements do not meet the requirements for safe operation and are assessed with insufficient buckling strength, then the methodology of this study could be applied. This work evaluates the case of buckling due to compressive forces on two parallel edges.

Before initiating any analysis, the on-site surveyor shall first examine whether the damage dealt to the plate is critical. For example, suppose its state is such that a patch repair is not beneficial to its structural integrity. In that case, a more significant repair at the area might be required, i.e., cropping and renewal. The engineer shall also investigate if the repair's design life is long-lasting or temporary, taking into account its installation and operating conditions. Specifically, in some instances where the environmental conditions might be harsh, i.e., high temperatures or high humidity, a patch might not be the optimal repair method. Finally, assuming the repair would take place while at sea, the available equipment (layers, lay-up techniques) onboard, as well as the deformation state (tensed/compressed plate), shall be considered. It should be noted that, even if the plate's shape is imperfect, i.e., has been deformed, the flexibility of the composite fabric allows for installation to be performed.

As previously stated, elastic buckling analysis is to be conducted using FEA. However, in a real-world problem, the conditions are not as ideal as is assumed by using this analysis foundation. A more accurate approach would be to include non-linearities in the modeling since the stiffness matrix changes during load application. When a plate stiffened across two parallel edges is subjected to a uniaxial compressive force along its unsupported edges, the internal stresses do not distribute uniformly. Specifically, lower stresses develop at the plate's center, while their value rises moving further from this area, i.e., internal stresses are minimum at the center and maximum near the supported edges. However, the plate's strips closer to the supports have higher rigidity since they are stiffened compared to the unsupported center. As the compressive load gradually increases, the latter is the first to buckle, while areas away from the center buckle at a higher value. This non-linear phenomenon represents a plate's post-buckling capacity, proving that a stiffened panel buckles ultimately at a value greater than the calculated elastic force (Hughes O. [15]). Since the buckling strength obtained by a non-linear analysis is greater than the one using linear assumptions, the latter stands for a safe preliminary assessment method for a plate's buckling capacity. Additionally, the unified rules with respect to srength (URS)

also base their buckling evaluation on an elastic approach with corrections for inelastic phenomena. Thus, elastic buckling analysis is performed for the plate's model with appropriate corrections for inelastic buckling as per Johnson's parabola.

### 2.2. Buckling Theory

Assume a rectangular plate, bordered by parallel stiffeners, with length $a_m$, width $b_m$ and thickness $t_m$ (the subscript "m" denotes the metallic plate). Let the isotropic material used for this element have its Young's modulus of elasticity be denoted by $E_m$ and its Poisson's ratio by $\nu_m$. All edges are considered to be simply supported due to the stiffeners acting as boundary supports. The critical value of the elastic compressive stress against buckling acting on the plate's short edges is given by (Timoshenko S. [16]):

$$\sigma_{E,cr} = \frac{N_{x,cr}}{t_m} = K_{cr}\frac{\pi^2 E_m}{12(1-\nu_m^2)}\left(\frac{t_m}{b_m}\right)^2 \qquad for \quad \sigma_{E,cr} \leq \sigma_0/2 \qquad (1)$$

where

$$K_{cr} = \min\left(m\frac{b_m}{\alpha_m} + \frac{a_m}{b_m}\frac{n^2}{m}\right)^2 \qquad (2)$$

In Equation (1), $N_{x,cr}$ is the critical compressive force at the plate's short edges and $K_{cr}$ is the critical buckling coefficient. In Equation (2), $\alpha = a_m/b_m$ is the plate's aspect ratio. The values m and n are the number of half-waves developed across the longitudinal (length's) and transverse (width's) directions. The most probable Eigen-buckling form has one half-wave across its transverse direction, i.e., n = 1, since it has a lower critical buckling coefficient value, as proven by comparing $K_{cr,n}$ and $K_{cr,n+1}$. It should also be noted that, for each successorial wave-form m, $K_{cr}$ tends to the value of 4 (as shown in Figure 1). This can be used as an approximate value for long plates (a ≫ b) that tend to buckle in half-waves with a length equal to the plate's width, i.e., a long plate is subdivided into buckled square plates, as shown in Figure 2.

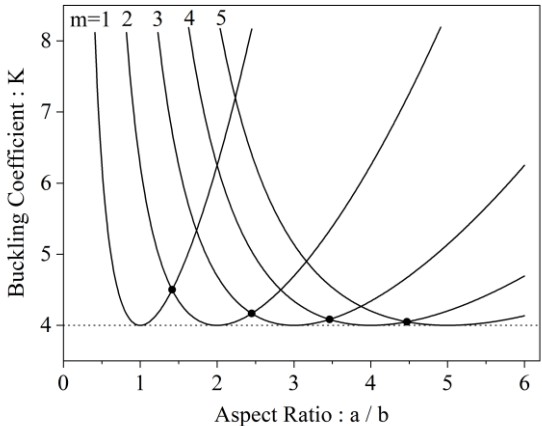

**Figure 1.** Compressive buckling coefficient curves for m = 1 through m = 5. Dots represent points of intersection between curves.

When using Equation (1), there is no limit to the critical stress's values, even if it surpasses the material's yield stress, not accounting for plasticity. Thus, Johnson–Ostenfeld's parabola for inelastic buckling is used while utilizing the elastic value obtained from Equation (1):

$$\sigma_{cr} = \sigma_{E,cr}, \text{ for } \sigma_{E,cr} \leq \sigma_y/2$$

$$\sigma_{cr} = \sigma_y\left(1 - \frac{\sigma_y}{4\sigma_{E,cr}}\right), \text{ for } \sigma_{E,cr} > \sigma_y/2 \qquad (3)$$

where $\sigma_y$ is the material's yield stress. According to Equation (1), the critical elastic buckling stress $\sigma_{E,cr}$ is analogous to the thickness squared and is inversely analogous to the width

squared. The critical buckling coefficient $K_{cr}$ is independent of the plate's thickness. Thus, if a 5% uniform thickness reduction due to corrosion (2.5% on each side) is applied, then a 10% reduction of the elastic buckling capacity occurs, a level of damage that the repair technique needs to recover.

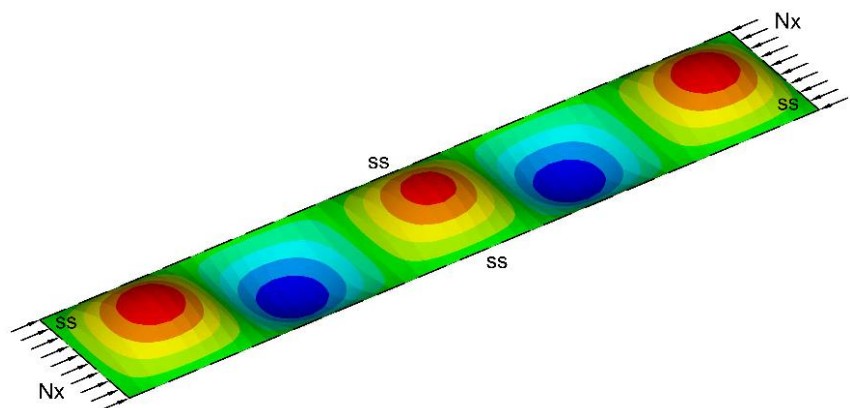

**Figure 2.** Long plate (a/b = 5) buckling eigen-form, representing the effect of buckling by squares for long plates.

### 2.3. Design Parameters

The stiffness and strength of a fiber-reinforced composite material are dependent on several more parameters apart from its length, width, and thickness. The most important are the ply orientation and stacking sequence, the matrix material, the fiber material, the fiber thickness (diameter) and the environmental conditions during fabrication, along with the fabrication method itself. When introducing its bonding with the metal substrate, the adhesive medium, installation conditions, and imperfections shall also be considered. In order to focus on the problem's main parameters, all other conditions are considered ideal, i.e., perfect bonding and absence of imperfections.

One of the main parameters is the patch's modulus of elasticity, i.e., the type of polymer used. Two cases were considered: a glass (GFRP) and a carbon (CFRP) plain weave composite fiber-reinforced polymer. However, in Section 2.5, it is proven through an OFAT (one-factor-at-a-time) statistical analysis that, the higher the modulus of elasticity, the better the structural response against compression. Hence, plain weave CFRP plies are used for the patch's laminate. Usually, the first ply attached to the adhesive is glass fiber for avoiding galvanic corrosion between the metal substrate and the carbon fiber.

An additional parameter is the patch's configuration for the repair. Two cases are considered: single-sided or double-sided patch (Figure 3). Once again, an OFAT analysis is performed for this problem; the buckling strength capacities for the same total number of plies were evaluated (see Section 2.5). This assessment provides evidence that a single strap has better results considering the rehabilitated plate's buckling capacity.

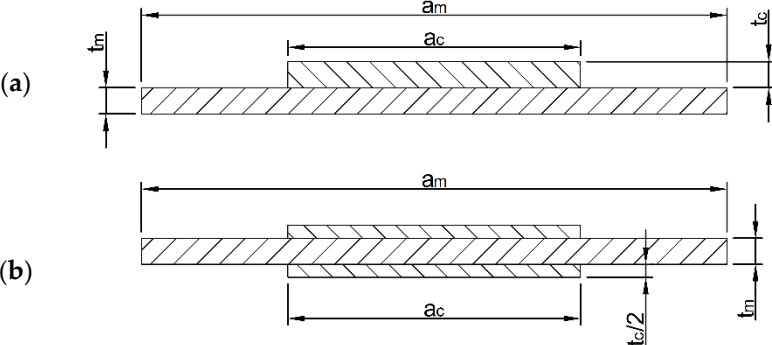

**Figure 3.** Single-sided (**a**) and double-sided (**b**) patch configuration.

Thus, assuming square shapes for both the plate and patch (to avoid differences between lengths and widths that would increase the problem's complexity), the design parameters are the patch's length and thickness (see Figure 4). The length's value is dependent on the metal substrate's dimensions; its range shall be taken as a percentage of that. Thus, if the square patch's respective value is denoted by $a_c$ (the subscript "c" corresponds to the composite patch), then its design space is $a_c = (0.1 \div 1) a_m$. As for the composite's thickness, it is solely dependent on each ply's thickness and the number of plies used. It is calculated as $t_c = N_{plies} t_{ply}$, where $N_{plies}$ is the number of plies and $t_{ply}$ stands for each individual ply's thickness. The range for the number of plies is $N_{plies} = (4 \div 32)$, while $t_{ply}$ is constant. For a plain weave CFRP ply $t_{ply} = 0.33$ mm.

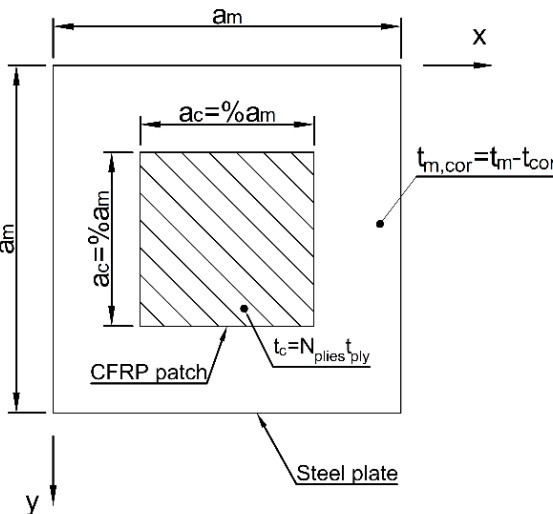

**Figure 4.** Schematic of plate's repair design with CFRP patch.

Having set the problem's inputs, the critical buckling stress is considered the main output. The objective of this method is to restore the plate's marginal buckling capacity reduced due to material wastage. A parameter necessary to set limit requirements for the repair technique's efficiency is the factor of safety (FoS), calculated as:

$$FoS = \sigma_{repaired} / \sigma_{intact} \qquad (4)$$

Results with a FoS $\geq 1$ are acceptable since the repaired structure's buckling strength is greater or equal to its original value.

### 2.4. Finite Element Model

The numerical simulations were performed using the commercial FE software ANSYS. The model consists of a metal steel plate with a CFRP patch fully connected (perfect continuity) and centrally placed with respect to the plate's surface. Since the only output parameter being examined is the eigen-buckling critical stress, the bonding between the composite and the metal substrate is considered ideal without modeling the stiffness of the adhesive or the resin-rich layer.

The plate is simply supported across all four of its boundary edges, simulating the existence of stiffeners. Due to the problem's symmetry, only one-quarter of the plate was modeled; corresponding boundary conditions are applied to the edges connecting with the rest of the geometry. Specifically, by taking a quarter of the geometry, as shown in Figure 5, the edges where the stiffeners are located are simply supported, so $U_Z = 0$ constraint is applied across these lines. At the edge connected to its adjacent quadrant and parallel to the x-axis, the degrees of freedom (DoF) $U_Y$, $ROT_X$ and $ROT_Z$ are restrained (equal to zero). Finally, on the edge parallel to the y-axis, $U_X$, $ROT_Y$, $ROT_Z = 0$ constraints are applied. The compressive load is applied as a unit force per unit length (equivalent to $N_x$ of Equation (1)).

By applying prestress to the model and solving the eigen-buckling static linear problem, the result given by the calculation is the critical buckling force $N_{x,cr}$. By dividing this value by the plate's thickness, the critical elastic buckling stress $\sigma_{E.cr}$ is obtained; corrections for inelastic buckling are implemented as per Equation (3), if applicable.

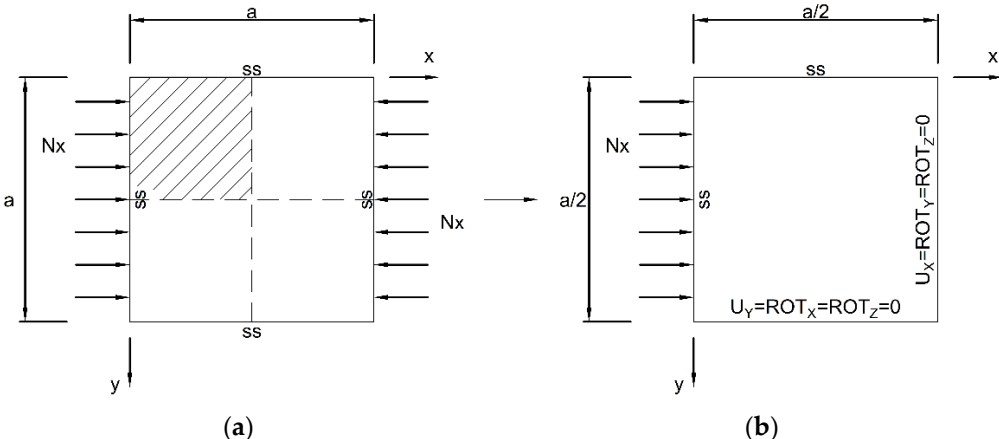

**(a)**                                        **(b)**

**Figure 5.** Plate's entire section model including loads and BCs (**a**) and the plate's shaded part quarter section model including loads and BCs (**b**).

A square plate with dimensions $a_m = b_m = 2500$ mm and thickness $t_m = 10$ mm has been considered in the analysis for illustrative purposes. The panel is characterized as a thin plate since $a_m/t_m \gg 20$; thus, shell elements are used for the virtual representation of both the plate and patch. In areas where the composite patch and plate overlap, the section's shell lay-up includes both instances, while, in other areas where the patch is absent, only the former is present. Two different shell element types can be used, namely, a four-node and an eight-node element with six degrees of freedom on each node, represented by SHELL181 and SHELL281 from ANSYS' library. These elements' efficiency, along with their size, is evaluated through a mesh convergence test. The geometry is test-meshed with an element size extending from 10 mm × 10 mm to 490 mm × 490 mm, with an increment of 10 mm (Figure 6). From this study, one may conclude that the assignment of an eight-node element SHELL281 and an element size equal to 50 mm × 50 mm provides accurate results. The meshed quarter model, with all assignments, loads, and boundary conditions (BCs), can be seen in Figure 7.

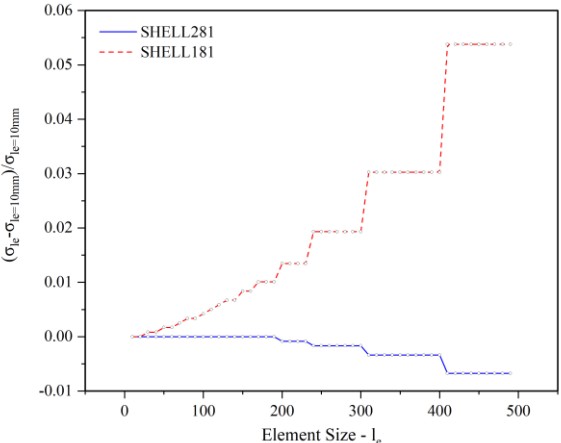

**Figure 6.** Mesh convergence for 4-node and 8-node elements.

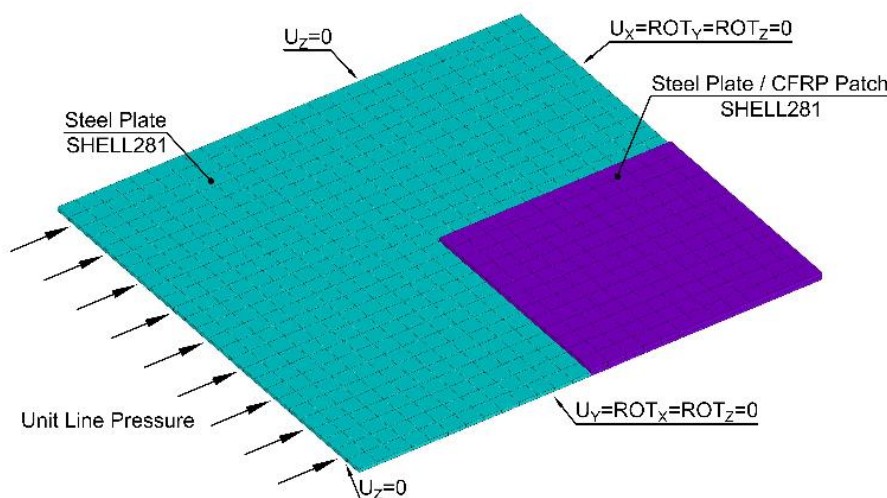

**Figure 7.** FE mesh of the repaired plate's quarter model with element/section assignments and boundary/loading conditions.

It should be noted that the mesh sensitivity study could have been avoided, since 8-node elements with a small size are eventually used. However, it is a common practice when simulating larger structures in FE programs to minimize calculation times and to maintain the model's accuracy through a mesh convergence test. Having performed the convergence test, the mesh attributes are then defined, i.e., element size and type. Since the current study's purpose is to introduce a methodology for solving the subject problem, this is a step that must be mentioned due to its significance in FEA problems in general.

The materials used are marine grade 'A' structural steel for the plate, which is an isotropic material with Young's modulus of elasticity $E_m = 206$ GPa, Poisson's ratio $\nu_m = 0.3$, and yield stress $\sigma_y = 235$ Mpa. The composite patch is a CFRP plain weave laminate that can be considered orthotropic, since there are three mutually perpendicular symmetry planes with respect to the alignment of the fibers (Kollar [17]). Its elasticity is $E_c = 42.95$ GPa across the longitudinal and transverse directions, while Poisson's ratio is $\nu_c = 0.3$.

### 2.5. OFAT Analyses

While analyzing the patch's design parameters in Section 2.3, the cases of the material's modulus of elasticity and the repair configuration were mentioned. In this paragraph, these two design parameters are examined through OFAT analyses to prove their effect on the structural stability of the rehabilitated plate and to demonstrate the reason that a single-sided CFRP patch was chosen as optimal for the repair design.

An OFAT analysis is commonly used in testing experimental factors one at a time, while all others are fixed, and is built on a statistical foundation. An argument could arise concerning the precision of such analysis, as design-of-experiments (DOE) exists as a robust alternative, offering analyses on several factors at once with their interactions. However, due to the simplicity of the design factors being tested, such complexity is unnecessary and is avoided.

The first factor is the modulus of elasticity, with two fiber materials being tested: carbon and glass. As aforementioned, a CFRP has a modulus of elasticity $E_{CFRP} = 42.95$ GPa while GFRP has $E_{GFRP} = 21.34$ GPa (Kollar [17]). The second factor is the repair's configuration: single-sided and double-sided patch (Figure 3). These factors were tested against the total number of plies used for the repair and the acquired FoS. The rest of the fixed design parameters are listed in Table 1, with the exception of the patch's length, which remains constant with a value of $a_c = 50\% \ a_m$.

**Table 1.** Plate's and patch's modeling particulars.

| Plate's Property | Symbol | Plate | Patch |
|---|---|---|---|
| Length | $a_m / a_c$ | 2500 mm | $[10\% \div 100\%]\ a_m$ |
| Thickness | $t_m / t_c$ | 10 mm | $N_{plies}\ t_{ply}$ |
| Corrosion | - | 5% | - |
| No. Plies | $N_{plies}$ | - | $4 \div 32$ |
| Ply Thickness | $t_{ply}$ | - | 0.33 mm |
| Material | - | Steel | CFRP |
| Young's modulus of elasticity | $E_m / E_c$ | 206 GPa | 42.95 GPa |
| Poisson's ratio | $\nu_m / \nu_c$ | 0.3 | 0.3 |

From the results shown in Figure 8, it is evident that the CFRP double-sided patch does not provide a superior stiffness compared to the single-sided that is of statistical significance. This evaluation is conducted on the basis of total buckling recovery (FoS = 1) with the least possible material usage, i.e., a minimum number of plies. Specifically, by comparing the repair configuration for each material, the single-sided patch shows better rehabilitation effects than the double-sided patch, with an increasing difference between them as the patch gets thicker. It is noted that the graph's x-axis shows the total number of plies, which means that each side of the double-sided patch consists of half of the number shown. A more significant difference in rehabilitation effects is apparent when comparing the two materials, with the CFRP exhibiting complete buckling capacity restoration for the least number of plies.

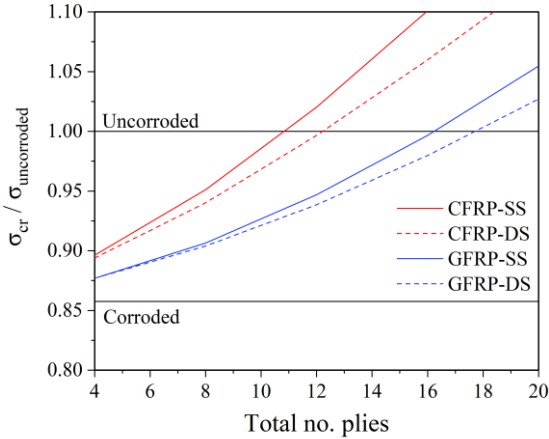

**Figure 8.** OFAT analysis for different patch's materials and configurations.

Thus, the CFRP single-sided patch configuration is chosen for this study. However, it should be noted that, when applying this technique in the industry, additional factors exist, in addition to the structural integrity. Some examples are the cost of materials, the properties of each material (e.g., curing, temperatures), and the equipment available at the time of the repair. Hence, in each case, the chosen configuration might not always be the one with the highest structural impact.

## 3. Design Assessment

### 3.1. Methodology

A scenario is assumed according to which, during a survey it is found that a plate has suffered uniform corrosion and safe operation is prohibited due to the amount of material wastage, i.e., thickness lost, in concerns of buckling occurring during operation (e.g., as shown in Figure 9). Let the repair method be that of a composite patch. The OFAT analysis (Section 2.5) proved that the best patch configuration for this type of repair is a single-sided CFRP patch, as it shows better rehabilitating structural behavior. After having these two primary variables set, the next step is choosing the patch's particulars for the case being

studied. The basis of assessing an option is the FoS, with a minimum requirement of FoS $\geq$ 1, although higher values might be requested by the owner or Classification Society. In order to select the design parameters, i.e., patch's length and number of plies, a design space is defined by employing DoE techniques. Specifically, a response surface (using response surface methodology or RSM) is constructed.

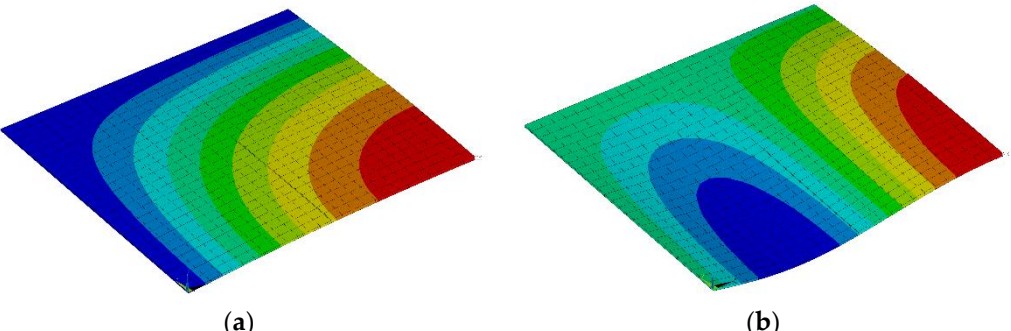

|  (**a**)  |  (**b**)  |

**Figure 9.** 1st (**a**) and 2nd (**b**) eigenmodes for the rehabilitated plate's quarter model.

RSM is a statistics tool whose function is to generate a map of response that utilizes the results of a two-level design, with a center point to detect the presence of curvature. A central composite design (CCD) is used to choose the surface's construction points, specifically a face-centered design (CCF). A CCD employs three types of points: factorial (represent main effects and show the two-factor interactions), center (show curvature and pure error), and star (show pure quadratic effects), as illustrated in Figure 10. The DOE's levels are three: low level (−1), center (0), and high level (+1). Each of these nine design points generates an output of interest, which, in the case being studied, is the FoS. These outputs are used to fit a quadratic surface over the design space, and, thus, the response surface is generated.

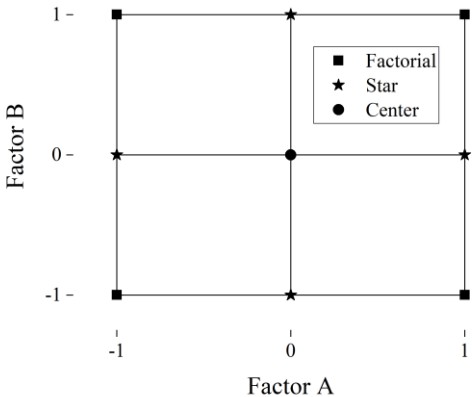

**Figure 10.** CCF design points.

Once the design space is generated, it is split between two regions: acceptable and unacceptable designs. This distinction is performed by having the required FoS as the evaluation tool. Thus, for FoS above the required value, any design parameters' combination is permitted. The final choice of design parameters is also dependent on several other factors, such as weight, cost, and time.

*3.2. Example*

In order to better understand the design methodology, the following example is given. Assume a plate with particulars listed in Table 1, i.e., a steel square plate of length 2.5 m and thickness 10 mm. During a survey, this plate is measured with a thickness of 9.5 mm, which is equivalent to 5% material wastage and is assessed to have a buckling risk if left untreated. Due to these concerns, a composite patch repair design is chosen for the

plate's buckling capacity rehabilitation. Specifically, a CFRP single-sided patch is chosen to be used, with available options for its length ($x_1$) between 10% and 100% of the plate's respective dimension and a number of plies ($x_2$) between 4 and 32. The designer has also set an additional FoS to that of 1, equal to 1.2. At this point, the response surface is generated for the given design parameters' range (listed in Table 2). The generated surface is plotted in Figure 11a, and the equation it is subjected to is:

$$FoS = 1.0405 - 0.3287x_1 - 0.0257x_2 + 0.0924x_1^2 + 0.0618x_1x_2 + 0.00057x_2^2 \qquad (5)$$

**Table 2.** CCD design points.

| Factor | Name | Unit | Low Level (−) | Center (0) | High Level (+) |
|--------|------|------|---------------|------------|----------------|
| $x_1$ | Length | mm | 10% $a_{plate}$ | 55% $a_{plate}$ | 100% $a_{plate}$ |
| $x_2$ | No. plies | - | 4 | 18 | 32 |

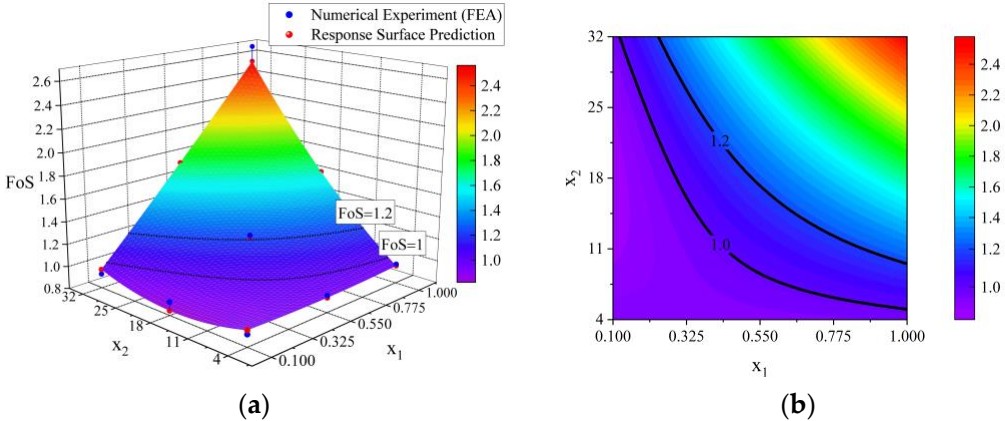

**Figure 11.** Response surface (buckling FoS) in a 3D design space (**a**) and its isolines projected to a 2D contour (**b**).

From Figure 11a, it can be seen that, for a patch covering the entire plate and with the maximum (32) number of plies, the obtained FoS is almost 2.6, far surpassing the FoS requirement. It should be noted that there was no need for corrections per Johnson's parabola according to Equation (3). In order to better visualize the acceptable combinations of the design parameters, Figure 11b is plotted as the top-down view of the surface, i.e., the view with the design parameters across its main axes. The designer may choose any length and number of plies combination above the FoS of his choice and calculate the obtained FoS using Equation (5).

In Figure 11a, the numerically calculated data using FEA and those obtained from using the mathematical model (i.e., Equation (5)) for the CCD points were also plotted. It is evident that most calculated points do not deviate much from the experimental data. In order to evaluate the model, the buckling surface was tested against lack-of-fit. For this reason, additional mid-points to those already existing were calculated, listed in Table 3. For this analysis, the FoS calculated from the FEA model (experimental data) was compared to the response surface (mathematical model's data). It was found that the R-squared value obtained is 0.99, which represents a ~99% fit. A histogram (Figure 12) and a corresponding normal probability plot (Figure 13) were also plotted. These charts show whether the data points follow a normal distribution and whether deviating statistical noise exists in the model; by studying these graphs, one can deduct that it does not.

**Table 3.** Lack-of-fit test points.

| Factor | Name | D + E [1] | E [2] | D + E [1] | E [2] | D + E [1] |
|---|---|---|---|---|---|---|
| $x_1$ | Length | 10% $a_{plate}$ | 32.5% $a_{plate}$ | 55% $a_{plate}$ | 77.5% $a_{plate}$ | 100% $a_{plate}$ |
| $x_2$ | No. plies | 4 | 11 | 18 | 25 | 32 |

[1] Design and evaluation point, [2] Evaluation point.

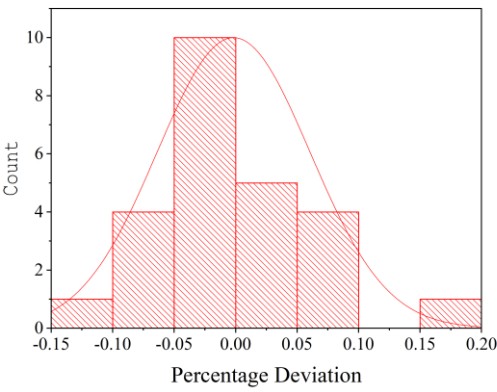

**Figure 12.** Histogram of the percentage deviation of the model's data and the experimental values.

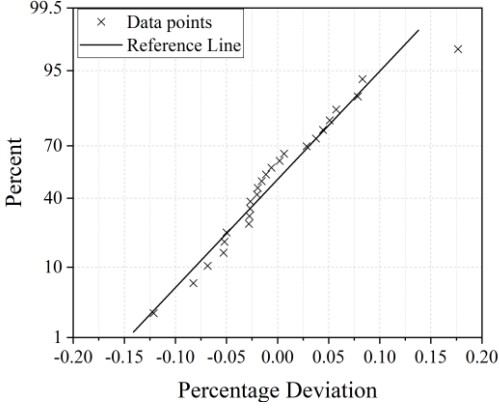

**Figure 13.** Normal probability plot of the percentage deviation of the model's data and the experimental values.

## 4. Conclusions

This study was focused on developing a new repair method for corroded marine metal plates susceptible to uniaxial buckling. By avoiding hot-work techniques, the application of composite patches can rehabilitate the plate's initial uncorroded buckling capabilities against the load mentioned above. The design parameters that were examined were the patch's length and thickness, assuming a square plate and patch. Additionally, non-linear parameters such as debonding and post-buckling strength were ignored. By employing design-of-experiments methods, specifically, response surface methodology, a mathematical model that predicts the acquired factor of safety for various combinations of the design parameters was constructed within the design space. It was possible to evaluate this model as an acceptable method for calculations through statistical analysis.

The results proved that a design methodology for publishing composite patch repair guidelines is possible. The same philosophy could be applied for other parameters, e.g., developing a model for different plate lengths and thicknesses, with the patch's variables fixed. However, additional design parameters should be employed to develop a more accurate model, i.e., non-linearities, debonding, fracture. The response surface would change or be limited to a stricter acceptable range by introducing these parameters, but the principles for designing the repair guidelines would remain the same. Additionally,

experimental results from lab testing are required to validate the method further and to evaluate its feasibility in real-world operations where environmental idealizations and other assumptions (such as neglecting material uncertainties) do not exist. A comparison between the models developed using numerical and experimental data would also be interesting. Finally, the method could be optimized according to other design requirements, such as minimizing added weight by developing a response surface for the patch's volume and finding its minimum value on a specified buckling FoS solution.

**Author Contributions:** Conceptualization, K.N.A.; Methodology, N.K. and K.N.A.; Formal analysis, N.K. and K.N.A.; Investigation, N.K.; Writing—original draft, N.K.; Writing—review and editing, K.N.A.; Validation, K.N.A. All authors have read and agreed to the published version of the manuscript.

**Funding:** This research received no external funding.

**Conflicts of Interest:** The authors declare no conflict of interest.

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
