# Peer review of "On the Design of Composite Patch Repair for Strengthening of Marine Plates Subjected to Compressive Loads"

_2673-3951, doi:10.3390/modelling3010009_

Round 1

Reviewer 1 Report

  • The aim of the paper is well introduced by explaining context of repairs.
  • The buckling background and theory are not too much detailed but it seems sufficient for the understanding.
  • OFAT acronym meaning should be given in page 5 and not page 8
  • DoE acronym should be given in page 8 and not page 10
  • Avoid to use 5 digits for thickness (0.33125mm)
  • Figure 9, sigma_cr should be used instead of sigma
  • What is the objective of the mesh sensitivity study if 8-nodes elements are used?
  • A smart design approach is presented but to be confirmed by comparing results with experiments. Or at least deviation due to materials uncertainties are to be considered in the approach.
  • Next steps should be clearly identified
  • Experimental data should be replaced by numerical data

Author Response

Please find our responses to the Reviewer's 1 comments in the attached file.

Reviewer 2 Report

The manuscript "On the design of composite patch repair for strengthening of marine plates subjected to compressive loads" is well-written and the methodology is sound.

The number of references on the topic is considered too low, therefore the introduction could be expanded.

A few aspects were misleading:

  • The methodology describes a method using composite material patches for metal structures. In the abstract and introduction, it was at first misleading that the topic has Composites and then the following is states "This phenomenon could lead to thickness reduction". The authors should make the transition clearer, and also address that for composite materials in fact the opposite is true -- fiber-reinforced polymer composite undergo hygroscopic swelling. [1] Analysis and Performance of Fiber Composites, 2nd ed.; John Wiley & Sons, Inc.: Hoboken, NJ, USA, 1990; ISBN 978-0-4715-1152-6.; [2] Prediction of Orthotropic Hygroscopic Swelling of Fiber-Reinforced Composites from Isotropic Swelling of Matrix Polymer. J. Compos. Sci. 2019, 3, 10, doi: 10.3390/jcs3010010; [3] Sinchuk, Y.; Pannier, Y.; Gueguen, M.; Tandiang, D.; Gigliotti, M. Computed-tomography based modeling and simulation of moisture diffusion and induced swelling in textile composite materials. Int. J. Solids Struct. 2018, 154, 88–96.
  • In Addition, what makes the statement more misleading is that lately, also fiber-reinforced composites have been gaining significant attention as structural materials in the aforementioned applications, not only steels. [4] Salama, M.M., et al., The first offshore field installation for a composite riser joint. Offshore technology conference [5] Multiscale Modelling of Environmental Degradation—First Steps. DOI: 10.1007/978-3-319-65145-3_8; [6] Integrating durability in marine composite certification. P. Davies, D.S. Rajapakse Yapa (Eds.), Durability of composites in a marine Environment, Springer Netherlands, Amsterdam (2014), pp. 179-194; [7] T. Brown. The impact of composites on future deepwater riser configurations MCE deepwater development conference (2017) [Amsterdam].

After the uncertainties in the story can be fixed, I recommend this manuscript for publication.

Author Response

Please find the responses to the Reviewer's 2 comments in the attached. 
